# Enhancing Minority Classes by Mixing: An Adaptive Optimal Transport Approach for Long-tailed Classification

**Jintong Gao[1], He Zhao[2], Zhuo Li[3,4], Dandan Guo[1]***

[1]School of Artificial Intelligence, Jilin University  [2]CSIRO's Data61
[3]Shenzhen Research Institute of Big Data
[4]The Chinese University of Hong Kong, Shenzhen
gaojt20@mails.jlu.edu.cn    he.zhao@ieee.org
221019088@link.cuhk.edu.cn    guodandan@jlu.edu.cn

## Abstract

Real-world data usually confronts severe class-imbalance problems, where several majority classes have a significantly larger presence in the training set than minority classes. One effective solution is using mixup-based methods to generate synthetic samples to enhance the presence of minority classes. Previous approaches mix the background images from the majority classes and foreground images from the minority classes in a random manner, which ignores the sample-level semantic similarity, possibly resulting in less reasonable or less useful images. In this work, we propose an adaptive image-mixing method based on optimal transport (OT) to incorporate both class-level and sample-level information, which is able to generate semantically reasonable and meaningful mixed images for minority classes. Due to its flexibility, our method can be combined with existing long-tailed classification methods to enhance their performance and it can also serve as a general data augmentation method for balanced datasets. Extensive experiments indicate that our method achieves effective performance for long-tailed classification tasks. The code is available at https://github.com/JintongGao/Enhancing-Minority-Classes-by-Mixing.

## 1 Introduction

Large-scale balanced datasets play a vital role in the remarkable success of deep neural networks for various tasks. However, datasets in many real-world applications often exhibit unexpected long-tailed distributions where most of the data belongs to several majority classes while the rest spreads across lots of minority classes. The model trained on such a long-tailed dataset will be biased toward majority classes, leading to poor generalizations about minority classes.

Re-weighting [1, 2, 3, 4, 5, 6, 7], over-sampling [8, 9, 10, 11, 12], under-sampling [13, 14, 15, 16], data augmentation [17, 18, 19, 20, 21, 22, 23], two-stage methods [24, 25, 26, 27], and other methods [28, 29] are common solutions to the long-tailed problem. Among them, over-sampling and data augmentation aim to balance the data distribution by oversampling or generating closely related minority classes. As the representative data augmentation techniques, mixup and its variants [30, 31, 32] have performed satisfactorily in the computer vision fields, whose key idea is constructing mixed samples by performing linear interpolations between data/features and corresponding labels. Considering mixup-based methods are designed for balanced data, applying them to long-tailed classification without any adjustments may ignore the specificity of long-tailed data distribution. Recently, some novel mixup-based methods for the long-tailed problem have been explored [22, 33,

---

*Corresponding author. This work was supported by NSFC (62306125).

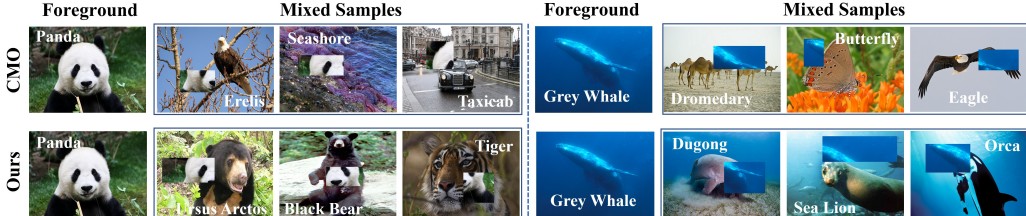

Figure 1: Visualization of mixed samples created by CMO and ours(OTmix), respectively.

34, 21]. As a representative example, Context-rich Minority Oversampling (CMO) [20] generates minority-centric images by mixing the background and foreground images, where background images are mainly from the majority classes while foregrounds are mainly from minority classes. The critical concept of CMO is to paste an image from a minority class (foreground) onto rich-context images from a majority class (background). Despite the effectiveness of CMO, it cannot be guaranteed that all the generated images are realistic and reasonable by randomly mixing the background and foreground images. For example, we visualize mixed samples created by ours and CMO in Fig. 1, where we take "Panda" and "Grey Whale" as two foreground images. CMO generates unreasonable mixed samples since pandas are hardly seen in the sky, the beaches, or the streets, and grey whales rarely appear in the desert, the flowers, or the sky. Therefore, instead of arbitrarily pairing the foreground and background images, it is essential to consider the semantic distances between the foreground and background images to generate more meaningful and advantageous samples.

In this work, we propose a more adaptive image-mixing method based on optimal transport (OT), which is a powerful tool for computing the distance between two distributions under the form of multiple sampling [35]. We consider an empirical distribution $P_{\text{back}}$ over the background images and an empirical distribution $Q_{\text{fore}}$ over the foreground images, where the former are more likely from the majority and the latter from minority classes. To learn the semantic similarity between background and foreground images, we formulate the learning of the similarity as the OT problem between $P_{\text{back}}$ and $Q_{\text{fore}}$. In this way, we can view the learned transport plan as the similarity between foreground and background samples, which can be naturally used as guidance to select the most relevant background image for each foreground image. Due to the importance of cost function in learning the transport plan for OT, we further design the cost function using the sample-level representation and class-level label information. Hence, the rich context of the majority classes as background images can be better transferred to the minority classes as foreground images for providing an adaptive way to learn the similarity between them and generate more semantically meaningful mixed images. Interestingly, our proposed OTmix can not only be combined with existing long-tailed methods but also be used as a new image-mixing strategy for balanced datasets.

Our main contributions are summarized as follows: (1) We propose a novel image-mixing method for generating reasonably mixed samples for long-tailed classification, where we introduce a distribution over the background images mainly from majority classes and another distribution over the foreground images mainly from minority classes. (2) By minimizing the OT distance between these two distributions, we view the learned transport plan as similar in guiding the image-mixing process. (3) We design the cost function based on the feature information and confusion matrix to learn the desired transport plan. Extensive experiments demonstrate the effectiveness of our method for long-tailed classification and also balanced classification.

## 2   Related Work

**Data Processing**   Learning relatively balanced classes from the data perspective is an effective solution to the long-tailed problem [36], which can be roughly divided into over-sampling, under-sampling, and data augmentation. Since our proposed method is related to the over-sampling and data augmentation methods, we describe them in detail below. Over-sampling aims to emphasize the minority classes and increase the instance number of the minority classes [20, 8, 9]. For example, the classical Synthetic Minority Over-sampling Technique (SMOTE) [12] uses the interpolation between a given minority sample and its nearest minority neighbors to create new samples. Data augmentation is another way of data processing to compensate the minority classes by generating and synthesizing new samples [18, 23]. For example, Major-to-minor Translation (M2m) [17] defines an optimization phase to augment minority classes via translating samples from majority classes. The Meta Semantic Augmentation (MetaSAug) approach [24] is proposed to perform effective

semantic data augmentation by learning more meaningful class-wise covariance. Sample-Adaptive Feature Augmentation (SAFA) [19] aims to extract diverse and transferable semantic directions from majority classes, and adaptively translate minority-class features along extracted semantic directions for augmentation.

In addition, mixup-based methods [32, 37, 38, 30, 39, 31, 40, 41, 42], the effective data augmentation methods in balanced datasets, have been developed to solve the long-tailed problem recently and are closely related to our work. For example, Mixup Shifted Label-aware Smoothing Model (MiSLAS) [27] uses mixup in its Stage-1 training without any adjustments. Rebalanced Mixup (Remix) [21] provides a disproportionately higher weight to the minority class when mixing two samples. Uniform Mixup (UniMix) [22] adopts a minority-favored mixing factor to encourage more majority-minority pairs occurrence and is further combined with the Bayes Bias (Bayias) caused by the inconsistency of prior. Targeted copy-paste augmentation [43] aligned with the novel domain enhances out-of-domain robustness on long-tailed camera trap dataset. [44] improves minority class performance from synthetic data by contrasting night and day backgrounds. To mitigate the overfitting issue in subpopulation shift, Uncertainty-aware Mixup (UMIX) [33] adopts the sampling uncertainty to reweight the mixed samples. CMO [20] constructs mixed samples between the background images more likely from the majority classes and foreground images more likely from the minority classes. However, it ignores the semantic similarity between images by randomly mixing them. Different from CMO, we formulate the image-mixing problem as the OT problem between two distributions and use the learned transport plan as guidance to mix images, providing a general and adaptive image-mixing method for long-tailed learning.

**Loss Function Engineering**   Designing effective training objectives is another solution to fight against class imbalance [45, 7, 5]. Label-Distribution-Aware Margin Loss (LDAM) [1] is proposed from the view of the generalization error bound, and LDAM-DRW adopts a deferred class-level re-weighting method. Balanced Softmax (BALMS) [2] accommodates the label distribution shift between training and testing and proposes a Meta Sampler that learns to re-sample training set by meta-learning. Label Distribution Disentangling (LADE) [4] loss disentangles the source label distribution from the model prediction based on the optimal bound of the Donsker-Varadhan representation. Besides, Focal Loss [46] determines the weights for samples with the sample difficulty in the object detection task, and Influence-balanced Loss (IB) [3] assigns different weights to samples according to their influence on a decision boundary. Our method can be naturally combined with many loss functions.

**Other Methods**   Recently, two-stage algorithms have been proposed [1, 5, 24], such as MiSLAS [27]. Meanwhile, a Bilateral Branch Network (BBN) [25] unifies the representation and classifier learning stages to form a cumulative learning strategy. RoutIng Diverse Experts (RIDE) [28] uses multiple experts to reduce the variance and bias of the long-tailed classifier. In addition, some approaches employ meta-learning, such as MetaSAug [24] and BALMS [2]. Causal Norm (CN) [29] pinpoints the causal effect of momentum and extracts the unbiased direct impact of each instance.

**Optimal Transport**   As a powerful tool, OT has been applied to generative models [47, 48, 49, 50, 51], computer vision [52, 53, 7, 54, 55, 56, 57], text analysis [58, 59, 60, 61], and etc. To the best of our knowledge, there are still very limited works for the imbalanced classification by means of OT. For example, [7] proposes a re-weighting method based on OT, and [62] introduces Optimal Transport via Linear Mapping (OTLM) to perform the post-hoc correction. Different from them, ours falls into the data augmentation group by mixing images. Another recent work is Optimal Transport for OverSampling (OTOS) [63], which moves random points from a prior uniform distribution into that of minority class samples based on OT. However, our proposed OTmix introduces OT to guide the mix between majority and minority images, which leverages the rich context of the majority classes guided by OT. Alignmixup [38] adopts OT to align two images and interpolate between two sets of features in a standard classification task, which is distinct from ours in terms of task and technical detail.

## 3   Background

**Long-tailed Classification**   Given a training set $\mathcal{D} = \{(x_i, y_i)\}_{i=1}^{N}$ for a multi-class problem with $K$ classes, if each class $k$ contains $n_k$ samples, we have that $\sum_{k=1}^{K} n_k = N$. Without loss of generality, we can always assume $n_1 \geq n_2 \geq \cdots \geq n_K$ for the long-tailed problem. Denote

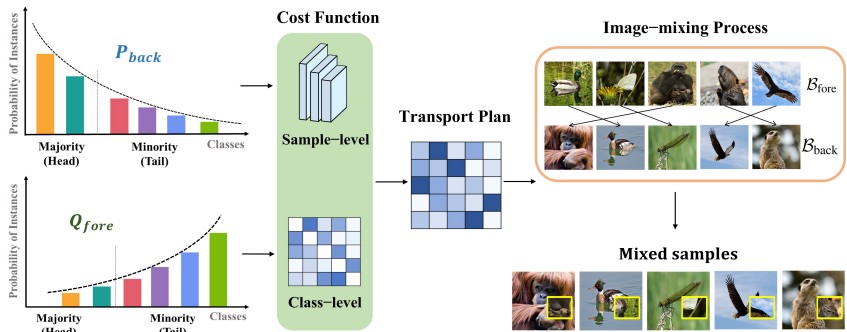

Figure 2: Adaptive image-mixing method with optimal transport.

the model parameterized with $\theta$ as $g(\theta)$, which is trained on $\mathcal{D}$ with the well-known Empirical Risk Minimization (ERM) algorithm [32] ignores such class imbalance and performs poorly on the minority classes [33].

**CutMix**    Here, we use the CutMix [37] to mix samples because of its simplicity and effectiveness, which replaces the image region with a patch from another training image. Denote $x \in \mathbb{R}^{W \times H \times C}$ and $y$ denote a training image and its label, respectively. CutMix combines two training samples $(x_i, y_i)$ and $(x_j, y_j)$ and generates the new sample $(\tilde{x}_{ij}, \tilde{y}_{ij})$ as follows:

$$\tilde{x}_{ij} = \mathbf{M} \odot x_i + (\mathbf{1} - \mathbf{M}) \odot x_j, \quad \tilde{y}_{ij} = \lambda y_i + (1 - \lambda)y_j, \tag{1}$$

where the combination ratio $\lambda$ is sampled from the beta distribution $\text{Beta}(\alpha, \alpha)$, $\odot$ is element-wise multiplication, and the binary mask $\mathbf{M} \in \{0, 1\}^{W \times H}$ indicates where to drop out and fill in from two images, $\mathbf{1}$ is a binary mask filled with ones. We provide more details about $\mathbf{M}$ in Appendix 7.1.

**Optimal Transport**    OT measures the minimal cost to transport between two probability distributions [35, 64, 65, 66, 67]. We only provide a brief introduction to OT for discrete distributions and refer the readers to [35] for more details. Denote two discrete probability distributions $p = \sum_{i=1}^{n} a_i \delta_{x_i}$ and $q = \sum_{j=1}^{m} b_j \delta_{y_j}$, where both $\boldsymbol{a}$ and $\boldsymbol{b}$ are discrete probability vectors summing to 1, $x_i$ and $y_j$ are the supports of the two distributions respectively, and $\delta$ is a Dirac function. Then the OT distance is formulated as follows: $\mathbf{OT}(p, q) = \min_{\mathbf{T} \in \Pi(p,q)} \langle \mathbf{T}, \mathbf{C} \rangle$, where $\mathbf{C} \in \mathbb{R}_{\geq 0}^{n \times m}$ is the cost matrix with element $C_{ij} = C(x_i, y_j)$ which reflects the cost between $x_i$ and $y_j$ and the transport probability matrix $\mathbf{T} \in \mathbb{R}_{\geq 0}^{n \times m}$ is subject to $\Pi(p, q) \coloneqq \left\{ \mathbf{T} \mid \sum_{i=1}^{n} T_{ij} = b_j, \sum_{j=1}^{m} T_{ij} = a_i \right\}$. The optimization problem above is often adapted to include a popular entropic regularization term $H = -\sum_{ij} T_{ij} \ln T_{ij}$ for reducing the computational cost, denoted as Sinkhorn algorithm [68].

# 4   Method

In this work, we propose a novel adaptive image-mixing data augmentation method based on OT for long-tailed classification, where the overall framework is shown in Fig. 2. We consider a distribution $P_{\text{back}}$ over the background images mainly from majority classes and another distribution $Q_{\text{fore}}$ over the foreground images mainly from minority classes. By minimizing the OT distance between these two distributions, we design the sample-level and class-level cost functions and use the learned transport plan as guidance for the image-mixing process to generate more semantically meaningful mixed samples between the two distributions.

**Main Objective**    Since majority classes usually have sufficient data and rich information, it is natural to leverage the majority classes to enhance minority classes. In this work, we aim to generate minority-centric images with majority contexts, combining the background and foreground images based on CutMix. The critical question is how to pair a background image and a foreground image. The **first factor** of the pairing strategy is that we would expect a background image to be more likely from the majority classes and a foreground image to be more likely from the minority classes, which has been modeled in CMO [20]. However, in addition to the first one, the **second factor** is the semantic similarity between the background and foreground images, which has not been studied in the literature and is the main focus of our paper. Specifically, suppose a set of candidate images

already satisfy the first factor, $i.e.$, background and foreground images are from majority and minority classes, respectively. Pairing these images arbitrarily may not guarantee that the combined images are semantically meaningful and helpful, $e.g.$, in Fig. 1.

In this paper, we aim to generate semantically meaningful images by proposing a pairing process that satisfies the first and second factors at the same time. Our proposal is derived from a distribution-matching perspective. First, we consider the distribution $P_{\text{back}}$ of the background images as the empirical distribution of all the images in the training dataset, which is a (discrete) uniform distribution:

$$P_{\text{back}} = \sum_{i=1}^{N} \frac{1}{N} \delta(x_i, y_i). \tag{2}$$

Although $P_{\text{back}}$ is uniformly distributed, because the majority classes have significantly more images than the minority classes, it is natural that the samples from $P_{\text{back}}$ are more likely to be images from the majority classes, which is what we expect. Secondly, we introduce the distribution over the foreground images, which is a discrete distribution over all the training images:

$$Q_{\text{fore}} = \sum_{j=1}^{N} w_j \delta(x_j, y_j), \tag{3}$$

where $w_j$ is the weight of the $j$-th sample. In this case, we expect that a sample from $Q_{\text{fore}}$ is more likely to be an image of the minority classes. Thus, $Q_{\text{fore}}$ cannot be a uniform distribution. Accordingly, we can specify $w_j$ in several ways, such as the (smoothed) inverse class frequency [69, 70, 71], or the adequate number of samples [72]. Taking the inverse class frequency as the example, we can sample the $k$-th class with the following probability: $q_k = \frac{1/n_k^r}{\sum_{k=1}^{K} 1/n_k^r}$, where $r > 0$ controls the smoothness of $q_k$, $r = 1$ indicates the inverse class frequency and $r = 1/2$ means the smoothed version, $i.e.$, if $r$ increases, the weight of the minority class becomes increasingly larger than that of the majority class. Finally, we can use $w_j = q_k \frac{1}{n_k}$ if $y_j = k$.

For now, with our construction of $P_{\text{back}}$ and $Q_{\text{fore}}$, the first factor is satisfied. For the second factor that requires the semantic similarity between a pair of background and foreground images, we formulate it as an optimization problem of the entropic OT:

$$\text{OT}_\epsilon(P_{\text{back}}, Q_{\text{fore}}) \overset{\text{def}}{=} \min_{T_{ij} \in \Pi(P_{\text{back}}, Q_{\text{fore}})} \sum_{i,j}^{N,N} C_{ij} T_{ij} - \epsilon \left[ \sum_{i,j}^{N,N} - T_{ij} \ln T_{ij} \right], \tag{4}$$

where $\epsilon > 0$ is the hyper-parameter for the entropic constraint, $C_{ij}$ is the transport cost function, and the transport probability $T_{ij}$ satisfies $\Pi(P_{\text{back}}, Q_{\text{fore}}) := \left\{ T_{ij} \mid \sum_{i=1}^{N} T_{ij} = w_j, \sum_{j=1}^{N} T_{ij} = \frac{1}{N} \right\}$. Notably, as an upper-bounded positive metric, $T_{ij}$ indicates the transport probability between the $i$-th background image and the $j$-th foreground image, which can be naturally used to measure the importance of each background image for the foreground image when performing image-mixing.

**Cost Function** $C_{ij}$ measures the distance between $(x_i, y_i)$ in $P_{\text{back}}$ and $(x_j, y_j)$ in $Q_{\text{fore}}$. As the main parameter for defining the transport distance between probability distributions, $C_{ij}$ plays an important role in learning the optimal transport plan, which can be flexibly defined in different ways. For clarity, we can reformulate the concerned model $g(\theta)$ as $g_2(g_1(\theta_1); \theta_2)$, where $g_1(\theta_1)$ denotes the feature extractor parameterized with $\theta_1$ and $g_2(\theta_2)$ is the classifier parameterized with $\theta_2$. We explore a few conceptually intuitive options of $C_{ij}$. A simple but straightforward way is defining the **sample-level** $C_{ij}$ with the features of $x_i$ and $x_j$:

$$C_{ij} = 1 - \cos(z_i, z_j), \tag{5}$$

where $\cos(\cdot, \cdot)$ is the cosine similarity, $z_i = g_1(x_i; \theta_1) \in \mathbb{R}^e$ and $z_j = g_1(x_j; \theta_1) \in \mathbb{R}^e$ denote the $e$-dimensional representation of $x_i$ and $x_j$, respectively. Now $C_{ij}$ will be small if two samples have similar or close features, where the cost is influenced by the feature extractor $\theta_1$. Besides, we can also define the **class-level** $C_{ij}$ only using the label information. Specifically, a normalized confusion matrix $\mathbf{F} \in \mathbb{R}_{\geq 0}^{K*K}$ can be estimated with an unbiased validation set or a small balanced subset sampled from $\mathcal{D}$. Summarizing the prediction results of the target model $g$, the confusion matrix $\mathbf{F}$ shows how the model $g$ is confused when it makes predictions. Since we focus on combining the foreground images mainly from minority classes and background images mainly from minority classes, we can set the diagonal element in $\mathbf{F}$ to 0, denoted as $\hat{\mathbf{F}}$, to avoid mixing the images from the same class. After that, we can define $C_{ij}$ with the ground-truth labels of two samples:

$$C_{ij} = 1 - \hat{F}_{y_i, y_j}, \tag{6}$$

where $\hat{F}_{y_i,y_j}$ indicates the element of row $y_i$, column $y_j$, of the matrix $\hat{\mathbf{F}}$. If $y_i$ and $y_j$ are from the same class, $\hat{F}_{y_i,y_j} = 0$ and $C_{ij} = 1$; if $y_i$ and $y_j$ are not from the same class but are easy get to confused by the model $g$, $\hat{F}_{y_i,y_j}$ would be large and $C_{ij}$ would be small. Now, $C_{ij}$ is affected by the feature extractor $\theta_1$ and classifier $\theta_2$. Intuitively, based on the features and labels of samples, the cost function can also be defined as:

$$C_{ij} = \omega(1 - \cos(z_i, z_j)) + (1 - \omega)(1 - \hat{F}_{y_i,y_j}), \tag{7}$$

where $\omega$ is a hyper-parameter for balancing the feature and label information and $C_{ij}$ will be small if two samples have similar features and are from two confused classes but $y_i \neq y_j$. Once the cost function $C_{ij}$ is defined, it can be fed into (4) for learning the transport plan $T_{ij}$ between foreground images and background images, where a small $C_{ij}$ tends to produce a large $T_{ij}$.

**Image-mixing Process**  By minimizing the OT problem, the resultant transport plan $\mathbf{T}$ provides an adaptive way to weigh the similarity between background and foreground images. Therefore, we can select the most suitable background image $x_{j'}$ for the foreground image $x_i$ as follows:

$$\arg \max_{j' \in N} j' = \{i \in N : j' = \max_{j \in N} T_{ij}\}. \tag{8}$$

We aim to generate new mixed samples by combining the foreground image $x_i$ and the most relevant background image $x_{j'}$ and their corresponding labels. As their name implies, $x_{j'}$ is used as a background image while $x_i$ provides the foreground patch, which can be pasted onto $x_{j'}$. Recalling CutMix in (1), the detailed mixing process is expressed as:

$$\tilde{x}_{ij'} = \mathbf{M} \odot x'_j + (\mathbf{1} - \mathbf{M}) \odot x_i, \qquad \tilde{y}_{ij'} = \lambda y'_j + (1 - \lambda)y_i. \tag{9}$$

**Training Details**  During the training process, we adopt a mini-batch setting to integrate our proposed OTmix with deep neural networks, where we learn $\mathbf{T}$ and $\theta$ alternatively. More specifically, at each iteration, we can sample a mini-batch $\mathcal{B}_{\text{fore}}$ from $Q_{\text{fore}}$ and another mini-batch $\mathcal{B}_{\text{back}}$ from $P_{\text{back}}$ to build $\hat{Q}_{\text{fore}}$ and $\hat{P}_{\text{back}}$, both of which have $M$ samples and $M < N$. In step (1), we can minimize the OT distance between $\hat{Q}_{\text{fore}}$ and $\hat{P}_{\text{back}}$ with (4) to learn the transport plan $\mathbf{T}$. In step (2), we can use the learned transport plan to select the most relevant background image $x'_j$ for each foreground image $x_i$ based on (8). In step (3), we can generate the mixed samples with (9) and minimize the classification loss to train the model $g$ parameterized by $\theta$:

$$L = \mathbb{E}_{(x_i,y_i)\sim\mathcal{B}_{\text{fore}}} \left[ \mathbb{E}_{(x'_j,y'_j)\sim\mathcal{B}_{\text{back}}} \left[ \ell\left(y_{ij'}, g(\tilde{x}_{ij'}, \theta)\right) \right] \right]. \tag{10}$$

Since the cost function relies on either the parameter $\theta_1$ in the feature extractor or the parameter $\theta$ in the whole model $g$, it might be inaccurate in the early training stage, resulting in an undesired transport plan matrix. Consequently, to avoid selecting the unsatisfactory background image for the foreground image and generating harmful mixed samples, it is more beneficial to randomly select $x'_j$ for $x_i$ in the early stage. To this end, we adopt $y_{\text{random}} \sim \text{Bernoulli}(\frac{t}{T})$ to decide whether randomly mixing the background and foreground pair, where $T$ is the number of training epochs and $t$ indicates the current epoch. In the early stage, such as $t \leq T/2$, we can get $y_{\text{random}} = 0$ with a high probability, where we randomly mix the background and foreground images like CMO [20]; in the late stage, such as $t > T/2$, we are more likely to sample $y_{\text{random}} = 1$, where we can learn $\mathbf{T}$ with (4) and select the most relevant background image for each foreground image. Moving beyond CMO, we introduce a more adaptive and elegant image-mixing data augmentation method for long-tailed classification. We summarize our proposed method in Algorithm 1 and highlight steps (1), (2), and (3).

**Using OTMix for Balanced Classifications**  In this case, our method can be viewed as a better alternative to other augmentation methods that mix images randomly [37, 30]. Applying OTMix in balanced classifications is similar to doing it in the imbalanced problems except that we construct both $P_{\text{back}}$ and $Q_{\text{fore}}$ as uniform distributions.

# 5  Experiments

## 5.1  Experimental Settings and Implementation Details

**Datasets**  Following [23, 25, 27, 20], we evaluate our method on long-tailed classification benchmark datasets: CIFAR-LT-10 [73], CIFAR-LT-100 [73], ImageNet-LT [74], and iNaturalist 2018 [75]. For

---
**Algorithm 1** Adaptive Image-mixing Method with Optimal Transport.
---
**Require:** Training dataset $\mathcal{D}$, model $g$ with parameter $\theta$, hyper-parameters $\{\omega, \alpha, \beta, \epsilon, r\}$;
1: Build discrete distributions $Q_{\text{fore}}$ with (3) and $P_{\text{back}}$ with (2);
2: **for** $t = 1, 2, ..., T$ **do**
3:     Sample mini-batch $\mathcal{B}_{\text{fore}}$ from $Q_{\text{fore}}$ and $\mathcal{B}_{\text{back}}$ from $P_{\text{back}}$; Build $\hat{Q}_{\text{fore}}$ with $\mathcal{B}_{\text{fore}}$ and $\hat{P}_{\text{back}}$ with $\mathcal{B}_{\text{back}}$;
4:     Sample $y_{\text{random}} \sim \text{Bernoulli}(\frac{t}{T})$ ;
5:     **if** $y_{\text{random}} = 0$ **then**
6:         Generate the mixed samples with (1), where we randomly choose $x_j$ for $x_i$;
7:     **else**
8:         **Step (1):** Learn the transport plan $\mathbf{T}$ by minimizing $\text{OT}_\epsilon(\hat{Q}_{\text{fore}}, \hat{P}_{\text{back}})$ with (4);
9:         **Step (2):** Select the most relevant background image $x'_j$ for foreground image $x_i$ with (8);
10:        **Step (3):** Generate the mixed samples $(\widetilde{x}_{ij'}, y_{ij'})$ with (9);
11:    **end if**
12:    Update $\theta^{(t+1)} \leftarrow \theta^{(t)} - \beta \bigtriangledown_\theta L$ with (10);
13: **end for**
---

clarity, the imbalance factor is defined as the data point amount ratio between the most frequent and the least frequent classes, $i.e.$, $\varphi = \frac{n_K}{n_1}$. Among them, CIFAR-LT-10 (CIFAR-LT-100) are created from CIFAR-10 (CIFAR-100) [76] with $\varphi = \{100, 50, 10\}$, respectively. ImageNet-LT, containing 1,000 classes with $\varphi = 256$, is created from ImageNet [74]. Different from them, iNaturalist 2018 is a real-world and large-scale imbalanced dataset with 8,142 classes and $\varphi = 500$.

**Baseline Methods**   We consider several baseline methods: (1) Empirical risk minimization (ERM), training on the cross-entropy loss; (2) Mixup-based augmentation methods: Remix [21], Unimix [22], CMO [20]; (3) Other augmentation methods: M2m [17], Open-sampling [18], FSA [23], SAFA [19]; (4) Losses designed for long-tailed classifications: LDAM[1], DRW [1], BALMS [2], IB [3], LADE [4]; (6) Other methods: BBN [25], RIDE [28], Decouple [26], MisLAS [27].

**Implementation Details**   For all datasets, we use PyTorch [77] and SGD optimizer with momentum 0.9. Besides, we set $\omega$ for defining cost function in (7) as 0.05, $\epsilon$ for entropic constraint in (4) as 0.01, $r$ for the smoothness in (3) as 1, and $\alpha$ for sampling combination ratio as 4. For CIFAR-LT-10 and CIFAR-LT-100, we use ResNet-32 [78] as the backbone following [1] and use 240 epochs on a single GTX 2080Ti and set the initial learning rate as 0.1, which is decayed by 0.1, 0.1, and 0.01 at the 100th, 160th, and 200th epochs. For the ImageNet-LT, we employ ResNet-50 as the backbone following [20] and use 200 epochs on four GTX 2080Ti GPUs. The learning rate is initialized as 0.1 and decays by 0.1, 0.1, and 0.01 at the 40th, 80th, and 160th epochs. For the iNaturalist 2018, we use ResNet-50 as the backbone fol-

Table 1: Top-1 errors (%) of ResNet-32 on CIFAR-LT-10 and CIFAR-LT-100. "‡": our reproduced results. "†": results reported in the original paper.

| | CIFAR-LT-10 | | | CIFAR-LT-100 | | |
|---|---|---|---|---|---|---|
| | 100 | 50 | 10 | 100 | 50 | 10 |
| ERM‡ | 26.9 | 22.9 | 12.9 | 63.2 | 56.3 | 43.4 |
| LDAM‡[1] | 26.3 | 21.6 | 13.4 | 61.1 | 56.0 | 44.4 |
| ERM-DRW‡[1] | 24.3 | 18.9 | 11.9 | 58.0 | 54.3 | 41.8 |
| LDAM-DRW‡[1] | 23.0 | 19.1 | 11.8 | 57.4 | 52.2 | 45.0 |
| BALMS‡[2] | 22.7 | 19.1 | 11.8 | 58.0 | 53.1 | 41.6 |
| IB† [3] | 21.7 | 18.3 | 11.7 | 55.0 | 51.1 | 42.0 |
| LADE† [4] | – | – | – | 54.6 | 49.5 | 38.3 |
| BBN† [25] | 20.2 | 17.8 | 11.7 | 57.4 | 53.0 | 40.9 |
| RIDE (3 experts)‡ [28] | 18.4 | 16.0 | 13.7 | 51.4 | 48.6 | 40.2 |
| MiSLAS† [27] | 17.9 | 14.3 | 10.0 | 53.0 | 47.7 | **36.8** |
| ERM + Open-sampling† [18] | 22.4 | 18.2 | 10.6 | 59.7 | 55.2 | 41.9 |
| ERM + M2m† [17] | 21.7 | – | 12.1 | 57.1 | – | 41.8 |
| LDAM-DRW + SAFA† [1] | 19.5 | 16.4 | 11.1 | 54.0 | 50.0 | 40.9 |
| ERM + Remix† [21] | 24.6 | – | 11.8 | 58.1 | – | 40.6 |
| ERM + UniMix† [22] | 23.5 | – | – | 58.5 | – | – |
| ERM + CMO[20] | 25.0‡ | 18.6‡ | 11.5‡ | 56.1‡ | 51.7† | 40.5† |
| RIDE (3 experts) + CMO [20] | 17.8‡ | 15.4‡ | 12.1‡ | 50.0† | 47.0† | 39.8† |
| ERM + OTmix | 21.7 | 16.6 | 9.8 | 53.6 | 49.3 | 38.4 |
| LDAM + OTmix | 22.3 | 18.0 | 12.0 | 56.3 | 50.9 | 41.5 |
| **DRW + OTmix** | 16.9 | 13.8 | **9.4** | 52.0 | 47.4 | 37.3 |
| LDAM-DRW + OTmix | 18.2 | 16.0 | 11.8 | 52.0 | 47.6 | 41.0 |
| **BALMS + OTmix** | **16.0** | **13.5** | 9.8 | 53.2 | 47.7 | 37.7 |
| **RIDE (3 experts) + OTmix** | 17.3 | 14.8 | 11.3 | **49.3** | **46.2** | 39.2 |

lowing [1] and train 210 epochs on four Tesla A100 GPUs with an initial learning rate of 0.1, which is decayed by 0.1 in the 30th, 80th, 130th, and 180th epochs. Performances are mainly reported as the overall top-1 errors (%). Following [79], we also report the error rates on three disjoint subsets: many-shot classes with more than 100 training samples, medium-shot classes with 20 to 100 samples, and few-shot classes with 20 samples.

Table 2: Top-1 errors (%) of ResNet-50 on ImageNet-LT and iNaturalist 2018. "*": results reported in CMO. "†": results reported in origin paper.

| | ImageNet-LT | | | | iNaturalist 2018 | | | |
|---|---|---|---|---|---|---|---|---|
| | ALL | Many | Medium | Few | ALL | Many | Medium | Few |
| ERM* [20] | 58.4 | 36.0 | 66.2 | 94.2 | 39.0 | 26.1 | 36.5 | 44.5 |
| ERM-DRW[1] | 49.9* | 38.3* | 52.7* | 71.2* | 36.3† | – | – | – |
| LDAM-DRW* [1] | 50.2 | 39.6 | 53.1 | 69.3 | 30.0 | 30.0 | 29.8 | 30.1 |
| BALMS* [2] | 49.0 | 39.1 | 51.2 | 67.9 | 30.0 | 30.0 | 29.8 | 30.1 |
| IB† [3] | – | – | – | – | 34.6 | – | – | – |
| LADE* [4] | 48.1 | 37.7 | 50.7 | 68.8 | 30.0 | – | – | – |
| Decouple-cRT* [26] | 52.7 | 41.2 | 56.0 | 73.9 | 31.8 | 26.8 | 31.2 | 33.9 |
| Decouple-LWS* [26] | 52.3 | 42.9 | 54.8 | 70.7 | 30.5 | 29.0 | 30.2 | 31.8 |
| BBN* [25] | – | – | – | – | 30.4 | – | – | – |
| MiSLAS* [27] | 47.3 | – | – | – | 28.4 | 26.8 | 27.6 | 29.6 |
| RIDE (3 experts)* [28] | 45.1 | 33.8 | 48.3 | 65.1 | 27.8 | 29.8 | 27.8 | 27.3 |
| RIDE (4 experts)* [28] | 44.6 | 33.8 | 47.7 | 63.5 | 29.1 | 29.1 | 27.6 | 26.9 |
| ERM + M2m† [17] | 57.8 | – | – | – | – | – | – | – |
| Focal Loss+ FSA* [23] | – | – | – | – | 34.1 | – | – | – |
| LDAM-DRS + SAFA* [19] | – | – | – | – | 30.2 | – | – | – |
| ERM + Remix† [21] | 58.3 | – | – | – | 38.7 | – | – | – |
| ERM + CMO† | 50.9 | 33.0 | 57.7 | 79.5 | 31.1 | **23.1** | 30.7 | 33.4 |
| RIDE (3 experts) + CMO* [20] | 43.8 | 33.6 | 46.1 | 64.4 | 27.2 | 31.3 | 27.4 | 26.9 |
| **ERM + OTmix** | 48.0 | **30.0** | 54.1 | 77.7 | 30.5 | 30.7 | 29.5 | 31.6 |
| DRW + OTmix | 46.6 | 33.0 | 51.0 | 69.6 | 28.9 | 29.4 | 28.1 | 29.6 |
| LDAM-DRW + OTmix | 47.5 | 37.2 | 48.9 | 71.8 | 30.4 | 31.5 | 29.8 | 30.6 |
| BALMS + OTmix | 44.4 | 36.0 | 47.6 | 57.3 | 28.5 | 28.9 | 28.0 | 29.2 |
| **RIDE (3 experts) + OTmix** | **42.7** | 40.6 | **43.5** | **55.9** | **27.0** | 28.7 | **27.2** | **26.2** |

## 5.2 Experiments on Long-tailed Classification

We conduct experiments on CIFAR-LT-10 and CIFAR-LT-100 of ResNet-32 in Table 1, where OTmix+ERM achieves better performance than the mixup-based augmentation methods with ERM loss, especially for CMO+ERM. It confirms the validity of selecting a suitable pair to mix. Besides, OTmix with ERM loss produces comparable performance to complex long-tailed classification methods, such as specially designed losses and other augmentation methods. Considering OTmix can also be combined with some existing long-tailed losses and model architectures, we further combine it with LDAM-DRW loss [1], BALMS loss [2], and also the network with multiple experts (RIDE) [28]. It is clear that OTmix equipped with BALMS loss, DRW loss, and RIDE outperforms most of the competing methods except on CIFAR-LT-10 with $\varphi = 10$, where it is only weaker than MisLAS. These results illustrate the superiority of the proposed OTmix.

We further perform experiments on large-scale imbalanced ImageNet-LT and iNaturalist 2018 datasets. Table 2 demonstrates that OTmix based on ERM loss can still produce a comparable performance when compared with competing long-tailed losses, related mixup-based, and other data augmentation methods. Similarly, combining ours with other losses or model architecture can overall outperform these related baselines on ImageNet-LT and iNaturalist 2018, where **RIDE (3 experts) + OTmix** performs best. Furthermore, our method can improve the performance of the medium and few classes, which is the goal of long-tailed methods. We compare OTmix with mixup-based augmentation methods under various loss functions in Appendix 7.2 and computational cost in Appendix 7.4.

## 5.3 Experiments on Balanced Classification

To validate whether our proposed OTmix can be used for the balanced classification task, we adopt the commonly-used CIFAR-10 [76] and CIFAR-100 [76]. Notably, our proposed OTmix can be flexibly combined with the classical mixup-based methods when mixing images, where we consider Mixup [32], Cutmix [37], and SaliencyMix [30]. We train the networks using the same details as on the CIFAR-LT. At each training iteration, we randomly sample two mini-batches, *i.e.*,

Table 3: Top-1 errors (%) of our methods using ResNet-32 on balanced datasets.

| Method | CIFAR-10 | CIFAR-100 |
|---|---|---|
| **ResNet-32 (baseline)** | **11.8** | **34.2** |
| + Mixup [32] | 9.8 | 32.9 |
| **+ OTmix (Mixup)** | **8.1** | **32.3** |
| + Cutmix [37] | 8.6 | 31.6 |
| **+ OTmix (Cutmix)** | **5.8** | **28.0** |
| + SaliencyMix [30] | 7.2 | 31.2 |
| **+ OTmix (SaliencyMix)** | **5.1** | **21.7** |

$\mathcal{B}_{fore}$ and $\mathcal{B}_{back}$, from the balanced training dataset. Table 3 lists the experimental results on balanced CIFAR datasets, where OTmix produces better performance. It indicates the benefit of building a better image pair to perform image-mixing data augmentation for the balanced classification task. Moreover, **OTmix (SaliencyMix)** achieves the best top-1 error of 5.1% and 21.7% on CIFAR-10 and CIFAR-100 datasets, respectively. These results show that OTmix is still effective on balanced

classification tasks and can be combined with some advanced mixup-based methods, proving its flexibility and availability.

## 5.4 Analytical Experiments

Here we implement the ablation study under various settings on CIFAR-LT-100 with $\varphi = 100$ and summarize the results in Table 4. Our best-performing method uses the combined cost function with cosine similarity, sets the diagonal element in the confusion matrix $\mathbf{F}$ as 0, and relies on the maximum value in the transport plan matrix to select the background image for each foreground image. To reveal the influence of the cost function, we denote the cost function by only using the sample-level cost and the class-level cost as "Sample-level cost only" and "Class-level cost only", respectively. Besides, we also consider using euclidean distance rather than cosine similarity to define the cost function called "Euclidean distance". Compared with ours, which combines the sample-level and class-level costs and uses cosine similarity, all these variants perform slightly worse, demonstrating the benefit of combining features and labels to compute the cost function and the effectiveness of cosine similarity.

To avoid mixing the background and foreground images from the same category, we set the diagonal element in $\mathbf{F}$ as 0 and use the resultant $\hat{\mathbf{F}}$ to compute the cost function. Here, we consider its variant by calculating the cost function using the original confusion matrix $\mathbf{F}$, denoted as "Keeping same category", which can be found to outperform by OTmix. It shows that avoiding mixing samples from the same class is beneficial, no matter from minority or majority classes. Besides, we

Table 4: Ablation study on CIFAR-LT-100.

| Method | All | Many | Medium | Few |
|---|---|---|---|---|
| **ERM + OTmix** | **53.6** | **27.7** | **53.2** | **83.3** |
| Sample-level cost only | 55.1 | 30.8 | 56.1 | 85.5 |
| Class-level cost only | 54.5 | 30.1 | 55.9 | 84.5 |
| Euclidean distance | 54.1 | 27.9 | 54.0 | 84.8 |
| Keeping same category | 54.9 | 29.0 | 55.1 | 85.0 |
| Probability selection | 55.0 | 28.0 | 55.7 | 85.6 |
| Confusion matrix | 57.1 | 29.4 | 59.5 | 86.8 |

adopt the multinomial distribution, denoted as "Probability selection", to select the corresponding background image for the $i$-th foreground image, whose probability vector can be formulated as $\boldsymbol{p} = \text{softmax}(T_{i,1:M})$. We observe that selecting background images according to probability distribution is inferior to using the maximum value, where the latter chooses the most relevant background image for each foreground image mainly from minority classes without randomness. It proves the necessity of using the maximum value in the transport plan to build a pair for mixing.

Furthermore, we use the confusion matrix itself ("Confusion matrix" in Table 4) to design the image-mixing pair instead of OT. The confusion matrix can be from a pre-trained model on the imbalanced training set with standard ERM loss. Specifically, after sampling the background and foreground images, if the class $k_1$ of the foreground image is most easily confused with $k_2$ of the background image ($k_1 \neq k_2$), we use them to construct a pair. However, using the confusion matrix to build the image-mixing pair is inferior to ours. The possible reason is that OTmix provides a more adaptive way to measure the importance of each background image for the foreground image at each training iteration. We provide more analytical results in Appendix 7.3.

## 5.5 The Learned Transport Plan for Image-mixing

Recall that our method adaptively learns a transport plan between a batch of foreground images and a batch of background images. Fig. 3 shows our learned transport plan on CIFAR-LT-10 with $\varphi = 100$, whose list of class names is {airplane, automobile, bird, cat, deer, dog, frog, horse, ship, truck} from $k = 1$ to 10. To facilitate the visualization, we set the size of $\mathcal{B}_{\text{fore}}$ and $\mathcal{B}_{\text{back}}$ as $M = 10$, respectively. We observe that the learned transport plan can effectively capture the semantic correlations between foreground and background images. For example, the foreground image from the "truck" class (from the "ship" class) is highly related to the background image from the "automobile" class (from the "airplane" class). These correlations align with the confusion matrix in Fig. 6 of Appendix 7.3, however, OTmix is adaptive and specific to batches, while the confusion matrix is global and fixed. In addition to capturing the class-level similarity information, we can find that different background images from the same class have different importance to the same foreground image, $i.e.$, the sample-level similarity information. For instance, the transport

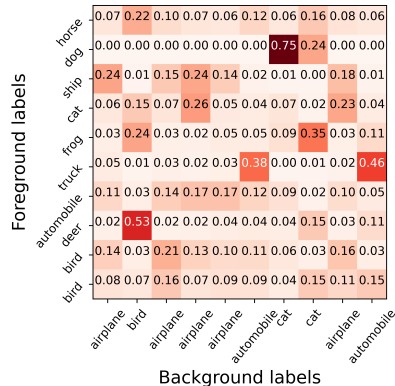

Figure 3: Our learned transport plan on CIFAR-LT-10 with $\varphi = 100$.

probabilities between the foreground dog image and two background cat images are $0.75$ and $0.24$, respectively. We attribute that to our designed cost function based on the sample-level feature and class-level confusion matrix when learning the transport plan. Therefore, a learned transport plan is an effective and adaptive way to guide the image-mixing process for long-tailed classification. We further compare the image-mixing statistical results of OTmix and CMO in Appendix 7.5 at one epoch, $i.e.$, the probability of each class in foreground distribution being mixed with each class in background distribution.

## 5.6 Visualization Results and Analysis

**T-SNE Visualization** To gain additional insight, we provide the T-SNE visualization of BALMS, BALMS+CMO, and BALMS+OTmix on CIFAR-LT-10 with $\varphi = 100$ in Fig. 4. Compared with BALMS+CMO, the majority class is more separable from the minority class2 in OTmix. Besides, the samples within the same class in our proposed method are usually closer than in CMO, $i.e.$, small intra-class. It indicates that our proposed method can generate more effective mixed samples to improve the long-tailed classification performance.

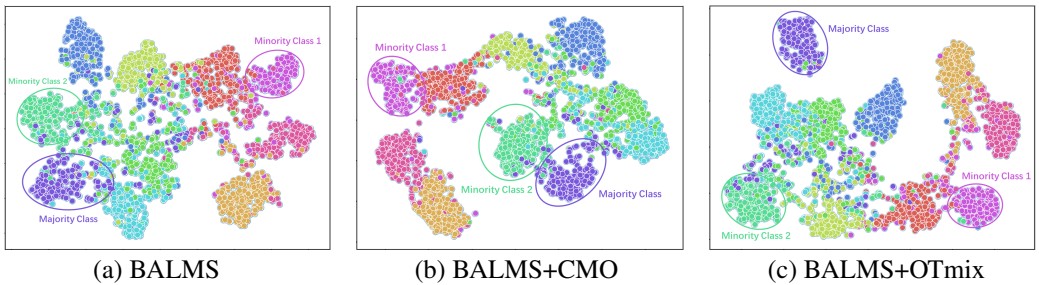

| (a) BALMS | (b) BALMS+CMO | (c) BALMS+OTmix |

Figure 4: T-SNE visualization of BALMS, BALMS+CMO, and BALMS+OTmix on CIFAR-LT-10.

**Mixed Image Pairs Visualization** To visually witness the intricate dynamics of the mixing process in our method, we provide the visualization results of image pairs for semantically meaningful mixed images on ImageNet-LT. As shown in Fig. 5, we plot the more semantically reasonable and relevant background image for each foreground image given the learned transport plan. For example, the semantic information of "Grey fox" is semantically relevant to "Dhole". It visually reveals that our method can generate meaningful and advantageous samples, benefiting the balanced and imbalanced classification tasks. More visualization results are available in Appendix 7.5.

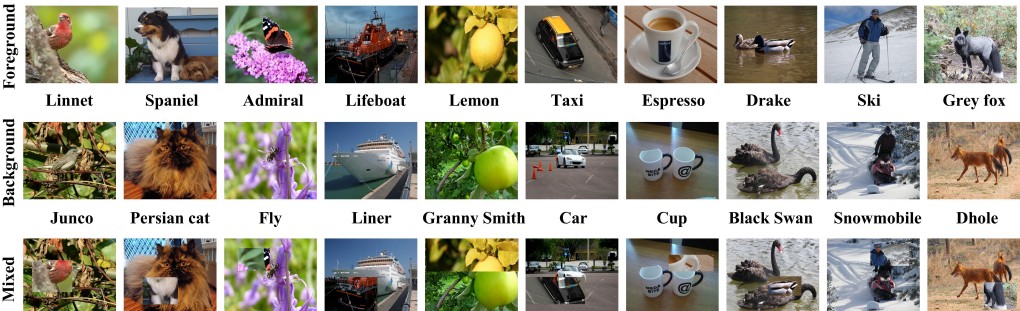

Figure 5: The visualization results of image pairs for meaningful mixed images on ImageNet-LT.

## 6 Conclusion

We have introduced a novel adaptive image-mixing augmentation method based on OT for long-tailed classification, with the goal of generating more semantically meaningful samples. We view the background images mainly from majority classes and the foreground images mostly from minority classes as two distributions, where we aim to minimize the OT distance between these two distributions. Moreover, we further design a combined cost function based on sample-level and label-level information. By viewing the learned transport plan as guidance to build a pair for image-mixing, we provide an effective way to weigh the background image for each foreground image. Our proposed OTmix has shown appealing properties that can be either combined with existing long-tailed methods or applied to balanced classifications. Experimental results validate that OTmix achieves competing performance on commonly long-tailed problems and balanced datasets.

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

# 7 Appendix

## 7.1 Details of CutMix

To sample the binary mask $\mathbf{M}$, we first sample the bounding box coordinates $\mathbf{B} = (r_x, r_y, r_w, r_h)$ indicating the cropping regions on $x_i$ and $x_j$. The region $\mathbf{B}$ in $x_i$ is removed and filled in with the patch cropped from $\mathbf{B}$ of $x_j$. In our experiments, we sample rectangular masks $\mathbf{M}$ whose aspect ratio is proportional to the original image. The box coordinates are uniformly sampled according to the:

$$
\begin{aligned}
r_x &\sim \text{Unif}(0, W), \quad r_w = W\sqrt{1-\lambda}, \\
r_y &\sim \text{Unif}(0, H), \quad r_h = H\sqrt{1-\lambda}
\end{aligned}
\tag{11}
$$

making the cropped area ratio $\frac{r_w r_h}{WH} = 1 - \lambda$. With the cropping region, the binary mask $\mathbf{M} \in \{0,1\}^{W \times H}$ is decided by filling with 0 within the bounding box $\mathbf{B}$, otherwise 1.

## 7.2 More Comparison Results

In this section, we conduct comparison studies with mixup-based long-tailed methods, including Remix [21], Unimix [22], and CMO [20] under various loss functions. We summarize the results on CIFAR-LT-10 and CIFAR-LT-100 in Table 6 and list the results on ImageNet-LT and iNaturalist 2018 in Table 5. We can find that our OTmix surpasses related mixup-based methods with varying loss functions and enhances the imbalanced classification. Besides, OTmix with different losses produces different classification performances due to their characteristics, where OTmix with BALMS can achieve better or competing performance than OTmix with other loss functions. These results reveal the effectiveness of our proposed method when combined with other loss functions.

Table 5: Top-1 errors (%) of mixup-based long-tailed methods under various loss functions on ImageNet-LT and iNaturalist 2018. "*": results reported in CMO. "†": results reported in origin paper.

| Method | | ImageNet-LT | | | | iNaturalist 2018 | | | |
|---|---|---|---|---|---|---|---|---|---|
| Mixup | Loss | ALL | Many | Medium | Few | ALL | Many | Medium | Few |
| None* | ERM | 58.4 | 36.0 | 66.2 | 94.2 | 39.0 | 26.1 | 36.5 | 44.5 |
| Remix† | ERM | 58.3 | – | – | – | 38.7 | – | – | – |
| CMO† | ERM | 50.9 | 33.0 | 57.7 | 79.5 | 31.1 | **23.1** | 30.7 | 33.4 |
| OTmix | ERM | **48.0** | **30.0** | **54.1** | **77.7** | **30.5** | 30.7 | **29.5** | **31.6** |
| None | ERM-DRW | 49.9* | 38.3* | 52.7* | 71.2* | 36.3† | – | – | – |
| Remix† | ERM-DRW | – | – | – | – | 29.5 | – | – | – |
| CMO† | ERM-DRW | 48.6 | 39.2 | 51.4 | **64.5** | 29.1 | 31.8 | 29.8 | **27.8** |
| OTmix | ERM-DRW | **46.6** | **33.0** | **51.0** | 69.6 | **28.9** | **29.4** | **28.1** | 29.6 |
| None* | LDAM-DRW | 50.2 | 39.6 | 53.1 | 69.3 | 30.0 | 30.0 | 29.8 | 30.1 |
| CMO† | LDAM-DRW | 48.9 | 38.0 | 52.6 | **69.2** | 30.9 | **24.7** | 30.5 | 32.7 |
| OTmix | LDAM-DRW | **47.5** | **37.2** | **48.9** | 71.8 | **30.4** | 31.5 | **29.8** | **30.6** |
| None* | BALMS | 49.0 | 39.1 | 51.2 | 67.9 | 30.0 | 30.0 | 29.8 | 30.1 |
| CMO† | BALMS | 47.7 | 38.0 | 50.9 | 63.3 | 29.1 | 31.2 | 30.0 | **27.7** |
| OTmix | BALMS | **44.4** | **36.0** | **47.6** | **57.3** | **28.5** | **28.9** | **28.0** | 29.2 |

## 7.3 More Analytical Results

**Confusion Matrix** To verify whether our method improves the performance of minority classes, we show the confusion matrices of ERM-DRW, ERM-DRW+CMO, and ERM-DRW+OTmix on CIFAR-LT-10 with $\varphi = 100$ in Fig. 6. We can find that ERM-DRW suffers a severe performance drop in the minority classes even though it can almost accurately predict the samples in the majority classes. ERM-DRW+CMO can improve the accuracy of the minority classes, which coincides with the statement of CMO. OTmix further enhances the generalization of minority classes and

Table 6: Top-1 (%) errors of mixup-based long-tailed methods with various loss functions on CIFAR-LT-10 and CIFAR-LT-100. "‡": our reproduced results. "†": results reported in the original paper.

| Method | | CIFAR-LT-10 | | | CIFAR-LT-100 | | |
|---|---|---|---|---|---|---|---|
| Mixup | Loss | 100 | 50 | 10 | 100 | 50 | 10 |
| None[‡] | ERM | 26.9 | 22.9 | 12.9 | 63.2 | 56.3 | 43.4 |
| CMO | ERM | 25.0[‡] | 18.6[‡] | 11.5[‡] | 56.1[†] | 51.7[†] | 40.5[†] |
| Remix[†] | ERM | 24.6 | – | 11.8 | 58.1 | – | 40.6 |
| UniMix[†] | ERM | 23.5 | – | – | 58.5 | – | – |
| **OTmix** | **ERM** | **21.7** | **16.6** | **9.8** | **53.6** | **49.3** | **38.4** |
| None[‡] | LDAM | 26.3 | 21.6 | 13.4 | 61.1 | 56.0 | 44.4 |
| CMO[‡] | LDAM | 25.9 | 22.7 | 13.1 | 58.0 | 54.3 | 44.1 |
| UniMix[†] | LDAM | 24.6 | – | – | 58.3 | – | – |
| **OTmix** | **LDAM** | **22.3** | **18.0** | **12.0** | **56.3** | **50.9** | **41.5** |
| None[‡] | ERM-DRW | 24.3 | 18.9 | 11.9 | 58.0 | 54.3 | 41.8 |
| Remix[†] | ERM-DRW | 20.2 | – | 11.0 | 53.2 | – | 38.8 |
| CMO | ERM-DRW | 19.5[‡] | 16.6[‡] | 11.3[‡] | 53.0[†] | 49.1[†] | 38.3[†] |
| **OTmix** | **ERM-DRW** | **16.9** | **13.8** | **9.4** | **52.0** | **47.4** | **37.3** |
| None[‡] | LDAM-DRW | 23.0 | 19.1 | 11.8 | 57.4 | 52.2 | 45.0 |
| Remix[†] | LDAM-DRW | 20.7 | – | 13.2 | 55.0 | – | 40.5 |
| CMO | LDAM-DRW | 19.0[‡] | 16.2[‡] | 12.4[‡] | 52.8[†] | 48.3[†] | 41.6[†] |
| **OTmix** | **LDAM-DRW** | **18.2** | **16.0** | **11.8** | **52.0** | **47.6** | **41.0** |
| None[‡] | BALMS | 22.7 | 19.1 | 11.8 | 58.0 | 53.1 | 41.6 |
| CMO | BALMS | 19.7[‡] | 15.9[‡] | 11.0[‡] | 53.4[†] | 48.6[†] | **37.7**[†] |
| **OTmix** | **BALMS** | **16.0** | **13.5** | **9.8** | **53.2** | **47.7** | **37.7** |
| None[‡] | Focal | 30.4 | 23.4 | 13.6 | 61.6 | 56.3 | 45.0 |
| CMO[‡] | Focal | 29.0 | 22.6 | 12.5 | 58.1 | 53.6 | 41.8 |
| **OTmix** | **Focal** | **27.0** | **20.9** | **12.0** | **57.8** | **53.3** | **40.6** |
| None[‡] | CB Softmax | 26.0 | 21.1 | 12.3 | – | – | – |
| CMO[‡] | CB Softmax | 25.7 | 20.2 | 12.2 | – | – | – |
| **OTmix** | **CB Softmax** | **24.0** | **18.8** | **12.0** | – | – | – |
| None[‡] | CB Sigmoid | 26.5 | 22.4 | 12.5 | – | – | – |
| CMO[‡] | CB Sigmoid | 27.4 | 20.8 | 12.1 | – | – | – |
| **OTmix** | **CB Sigmoid** | **24.9** | **20.6** | **11.8** | – | – | – |

maintains performance in majority classes, which thus outperforms the strong baselines on the overall performance. In Fig.7, we plot the classification results for each class on CIFAR-LT-10 with $\varphi = 100$, where we adopt the BALMS loss and LDAM loss, respectively. Compared with these baselines, OTmix provides a significant improvement in minority classes. We specifically note that the proposed method improves the accuracy over BALMS by 23% and over LDAM by 15% for the least frequent class 9 while degrading the accuracy for class 0 and class 1 by less than 2%. These results indicate that ours can achieve a more balanced classifier and ameliorate the generalization of minority classes.

**Discussion of Hyper-parameters and Training Settings**  To analyze the effect of different hyper-parameters and settings of OTmix, we conduct analytical experiments on CIFAR-LT-10 with $\varphi = 100$. The hyper-parameters include $\omega$, $\alpha$, and $r$, respectively. $\omega$ in (7) is employed to manage the degree of combination of the confusion matrix and feature information in Fig. 8(a). The best performance is achieved when $\omega = 0.05$, indicating the class-level cost based on the confusion matrix dominating the cost function. $\alpha$ affects the combined ratio in Cutmix in Appendix 7.1. As shown in Fig. 8(b), the larger the value of $\alpha$ is, the more complementary the distribution tends to be closer to a uniform distribution. The OTmix achieves an accuracy of 46.4% when $\alpha = 4$.

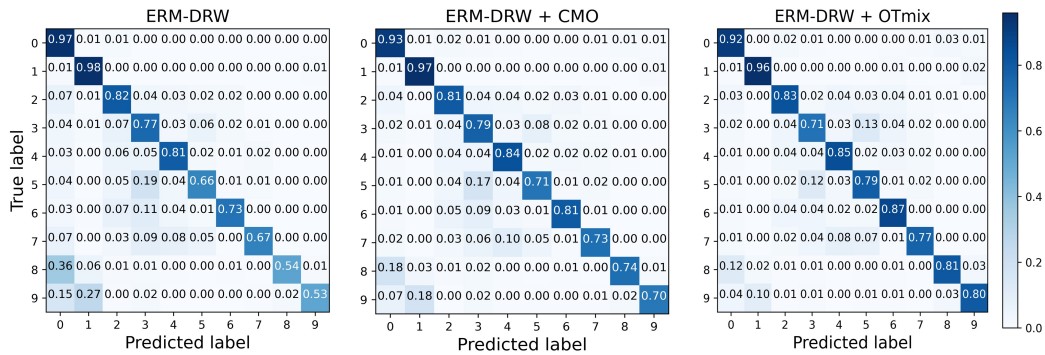

Figure 6: Confusion matrices of the ERM-DRW, ERM-DRW+CMO, and ERM-DRW+OTmix on CIFAR-LT-10 with $\varphi = 100$.

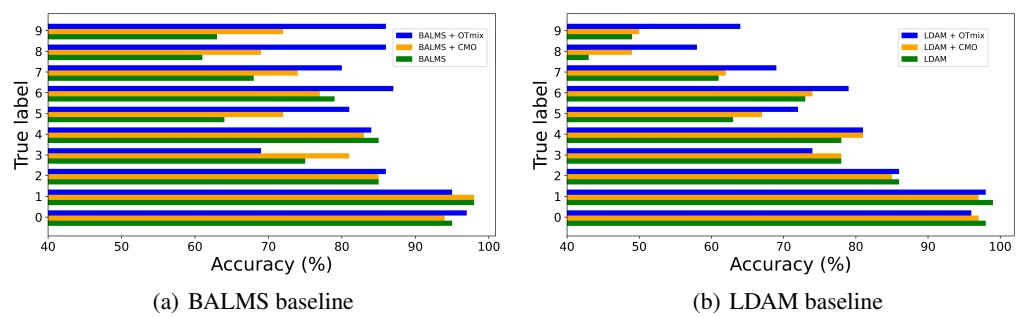

(a) BALMS baseline

(b) LDAM baseline

Figure 7: The classification results of different methods for each class on CIFAR-LT-10 with $\varphi = 100$, where (a) uses the BALMS loss and (b) adopts the LDAM loss. Class 0 stands for the majority class, and class 9 stands for the minority class.

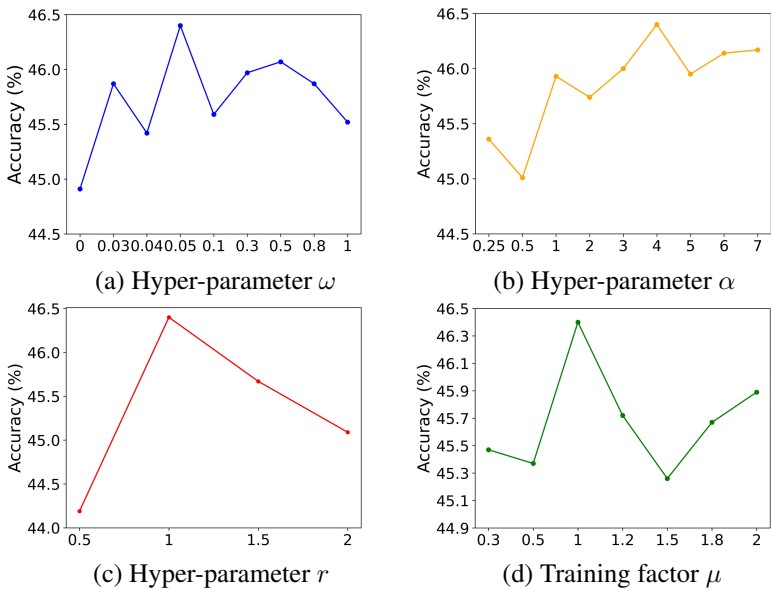

(a) Hyper-parameter $\omega$

(b) Hyper-parameter $\alpha$

(c) Hyper-parameter $r$

(d) Training factor $\mu$

Figure 8: Analytical experiments of different hyper-parameters and settings of the proposed method on CIFAR-LT-100 with $\varphi = 100$: (a-c) with various hyper-parameters, (d) with the setting of training factor.

The hyper-parameter $r$ controls the smoothness of $q_k$, which further decides the sample weight of foreground distribution in (4). As shown in Fig. 8(c), the smoothness factor $r = 1$ achieves the best performance, which indicates the inverse class frequency. Recalling that we adopt $y_{\text{random}} \sim$ Bernoulli$(\frac{t}{T})$ to decide whether mixing the background and foreground pair with our OTmix, where we use $\mu$ to set $y_{\text{random}} \sim$ Bernoulli$(\frac{t}{T})^{\mu}$ and explore the effect of $\mu$. As plotted in Fig. 8(d), ours with $\mu = 1$ produces the best performance. That is to say, we will randomly mix the background and foreground images with a high probability during the first half of training epochs, and we are more likely to use OTmix to mix the background and foreground pairs in the latter half of epochs.

**Mixed Fractions**   To examine the indispensability of the images unchanged in every batch, we conduct an additional experiment with the fraction of the mixed images in every batch on CIFAR-LT-10 and CIFAR-LT-100 under different methods and fractions. We denote $Fraction = \frac{Mixed}{Overall}$ in every batch. From the results in Table 7, when the number of mixed samples decreases ($Fraction \downarrow$), the performance of OTmix deteriorates significantly, which suggests that it is better to use mixed samples alone in every batch in our method.

Table 7: Classification errors of ResNet-32 on CIFAR-LT-10 and CIFAR-LT-100 under different methods and mixed fractions.

| Method | $Fraction$ | CIFAR-LT-10 | | | CIFAR-LT-100 | | |
|---|---|---|---|---|---|---|---|
| | | 100 | 50 | 10 | 100 | 50 | 10 |
| ERM | / | 26.9 | 22.9 | 12.9 | 63.2 | 56.3 | 43.4 |
| ERM + CMO | 100% | 25.0 | 18.6 | 11.5 | 56.1 | 51.7 | 40.5 |
| **ERM + OTmix** | **100%** | **21.7** | **16.6** | **9.8** | **53.6** | **49.3** | **38.4** |
| ERM + OTmix | 50% | 24.7 | 20.0 | 12.3 | 55.1 | 52.0 | 40.4 |
| ERM + OTmix | 25% | 23.6 | 20.1 | 11.1 | 58.1 | 53.1 | 40.5 |
| ERM + OTmix | 12.5% | 27.2 | 21.8 | 12.1 | 57.1 | 52.7 | 41.1 |

**Discussion of Confusion matrix**   Considering the confusion matrix played an important role in learning the cost function based class-level, we discuss the performance of our method and the computational cost of the normalized confusion matrix calculated on a balanced validation set, an imbalanced training set, and a small balanced subset sampled from imbalanced training set in Table 8. Meanwhile, the confusion matrix is represented as two states, fixed and adaptive. The former denotes where the confusion matrix remains unchanged in our approach and the latter changes dynamically in each epoch. From Table 8, we can draw a few observations: (1) With the same settings, the adaptive methods perform significantly better overall and few than the fixed methods. (2) Compared with the fixed balanced training setting $D_{fixed}^{bal}$, the fixed imbalanced training setting $D_{fixed}^{im}$ is preferable by providing more sample information ($D_{fixed}^{im} \gg D_{fixed}^{bal}$). However, large-scale samples can drastically increase the computational cost, making it difficult to implement adaptively. (3) Despite the time spent in the balanced training setting less, the balanced validation setting exhibits superior in terms of overall performance. To summarize, OTmix with the adaptively balanced validation setting enhances the suitability of OTmix for long-tailed classification.

Table 8: Classification errors of the ERM+OTmix with different confusion matrices under various methods and calculated settings on iNaturalist 2018.

| Method | Setting | ALL | Many | Medium | Few | Time |
|---|---|---|---|---|---|---|
| Fix | Balanced validation | **31.0** | **29.3** | **29.5** | 33.3 | 86s |
| Fix | Imbalanced training | 31.5 | 31.0 | 30.2 | **33.2** | 1277s |
| Fix | Balanced training | 31.8 | 32.7 | 30.3 | 33.5 | **58s** |
| Adaptive | Balanced validation (OTmix) | **30.5** | 30.7 | 29.5 | 31.6 | 86s |
| Adaptive | Balanced training | 31.1 | 32.9 | 30.6 | **31.1** | **58s** |

## 7.4 Computational Cost

The optimal transport (OT) problem in our method between probability distributions is computed by the Sinkhorn algorithm [68], which introduces the entropic regularization term for fast computation. To compute the OT distance between $n$ dimensional discrete distributions, the Sinkhorn algorithm requires the computational cost of $\mathcal{O}(n^2 log(n)/\varepsilon^2)$ reach $\varepsilon$-accuracy. In our case, $n$ corresponds to the batchsize, which is set to 128 in our experiments. We compare the computational cost of different methods on a Pentium PC with a single GTX 3060 GPU.

Table 9: Computational cost (s) per training epoch on long-tailed datasets.

| Method | CIFAR-LT-10 | CIFAR-LT-100 | ImageNet-LT | iNaturalist 2018 |
|---|---|---|---|---|
| ERM | 2.85 | 2.17 | 310.47 | 1251.29 |
| ERM+CMO | 3.35 | 2.85 | 319.73 | 1360.66 |
| ERM+OTmix | 4.59 | 3.79 | 333.64 | 1382.54 |

In addition, we also report the computational cost (s) per training epoch on long-tailed and balanced datasets, respectively. As shown in Table 9 and Table 10, mixup-based methods usually take more time than ERM. It is reasonable since mixup-based methods need to mix images. Besides, OTmix spends more time than CMO since we solve an OT problem to pair a background image and a foreground image. Still, introducing OTmix to existing mixup-based methods in balanced classification consumes more time. However, it is worth noticing that we only need to employ the OTmix during the late half phase of the training process. In summary, combining ours with others produces a better performance on long-tailed and balanced datasets with an acceptable cost.

Table 10: Computational cost (s) per training epoch on balanced datasets.

| Method | CIFAR-10 | CIFAR-100 |
|---|---|---|
| ERM | 11.9 | 12.8 |
| ERM+Mixup | 12.1 | 13.8 |
| ERM+OTmix (Mixup) | 17.1 | 18.1 |
| ERM+Cutmix | 13.2 | 15.7 |
| ERM+OTmix (Cutmix) | 19.1 | 21.5 |
| ERM+SaliencyMix | 14.2 | 17.9 |
| ERM+OTmix (SaliencyMix) | 21.5 | 24.9 |

## 7.5 More Visualization Results and Analysis

**Statistical Results of Mixed Images**    To intuitively reveal that OTmix is more effective than CMO, we show the statistical results of the mixed images generated by CMO and OTmix on CIFAR-LT-10, respectively. Specifically, we summarize the $10 \times 10$ matrix $m$ for the ten-class classification task in one training epoch, where element $m_{ij}$ denotes the number of pairs between the foreground images from the $i$-th class and the background images from the $j$-th class. As shown in Fig. 9(a), we can see that regardless of the foreground image from which class, CMO mainly mixes it with the background image from the majority classes. For example, the foreground images from the "truck" will be mixed with the background images from the "airplane" class, even if the truck is more similar to the automobile. However, Fig. 9(b) indicates that our proposed method builds more reasonable pairs by selecting the most relevant background image for each foreground image. For example, the "horse" is more easily confused with "deer" than "airplane". These results validate that ours can provide more reasonable generated samples than CMO.

**Visualization Results of Mixed Image Pairs**    To gain a more intuitive insight into the dynamic changes within the mixing process of our method, we provide more visualization results of mixed image pairs on iNaturalist 2018 in Fig. 10. The foreground image and the selected background image commonly have significant semantic similarity. It suggests that OTmix has the capacity to generate reasonably mixed samples for long-tailed classification.

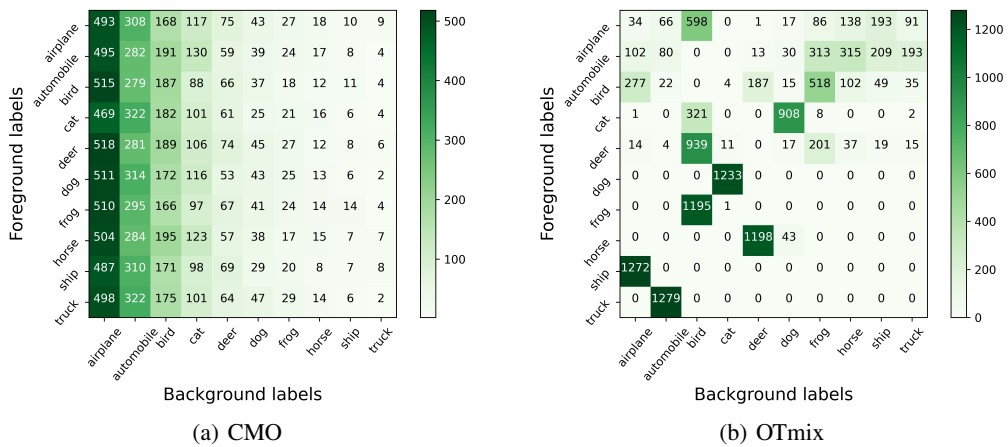

(a) CMO            (b) OTmix

Figure 9: Image-mixing statistical results of OTmix and CMO on CIFAR-LT-10 with $\varphi = 100$.

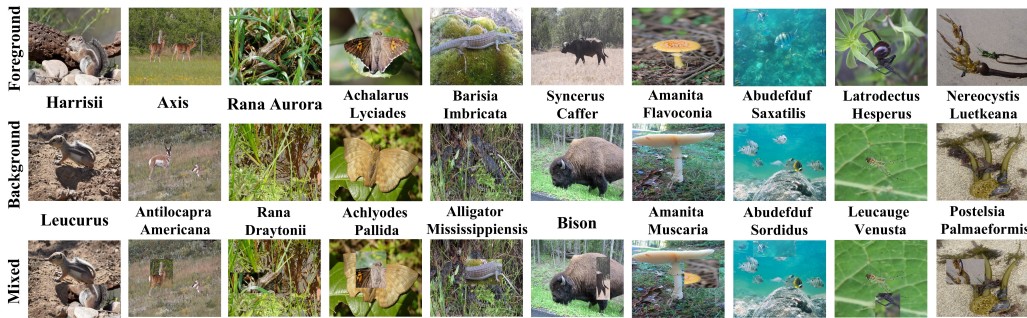

Figure 10: The visualization results of image pairs used to semantically meaningful mixed images on iNaturalist 2018.

## 7.6 Negative Societal Impacts and Limitations

This work develops a simple and effective image-mixing method for long-tailed learning, which has the potential to encourage researchers to derive new and better methods for the line of mixing images or long-tailed learning. However, if there is a sufficiently malicious or ill-informed choice of a long-tailed classification task or an image-mixing task, it may indirectly lead to a negative impact. Employing an imprecise or incorrect confusion matrix can mislead our method to build unsatisfactory pairs for image-mixing.

