# OpenReview forum: "Enhancing Minority Classes by Mixing: An Adaptative Optimal Transport Approach for Long-tailed Classification"
_NeurIPS.cc/2023/Conference — NeurIPS 2023 poster_

### Official Review · Reviewer_mXEg · 2023-07-05

**Soundness:** 3 good
**Presentation:** 3 good
**Contribution:** 3 good
**Rating:** 5
**Confidence:** 5

**Summary:**

This paper aim to solve long-tailed problem in the perspective of data augmentation. Specifically, the backgrounds of majority classes are mix with the foreground of minority classes. This work found out that randomly mixup images might face semantic misalignment, which impedes the effectiveness of this scheme. Therefore, this work use optimal transport to pair images with similar semantics. The paired images are mixed to form new training samples. On several benchmarks, this approach obtain good performance.

**Strengths:**

This paper has following strengths:

- This paper identity a issue of former mixup-based long-tailed method that the mixed samples are semantically mismatched. This paper proposes a sound solution to tackle this issue.

- This paper is easy to follow.

- This method can be unifed with former LT methods as a plug-in module.

- This paper has a clear motivation.

**Weaknesses:**

This paper has following weaknesses:

- The performance gain on RIDE (3 experts) compared with CMO is limited.

- The contribution is limited since it's a updated version of CMO.

- The key point is to find semantical similar pairs, it's crucial to visualize some pair results.

- OT is to calculate optimal plan for aligning distributions. But the ideal solution for a minority sample is to find its cloest minority neighbor. Does using OT also lead to a suboptimal solution?

**Questions:**

The complexity of estimate $T$

**Limitations:**

---

> ### Author Rebuttal · Authors · 2023-08-09
>
> **(1) Response to performance gain:** We would like to emphasize that in the highly competitive area of imbalanced classification, our performance gain over CMO is consistent and significant. Specifically, on CIFAR-LT-10 and CIFAR-LT-100 (imbalance factor=100,50,10), and ImageNet-LT (imbalance factor=256), the performance improvements of our method+RIDE over CMO+RIDE are encouraging, i.e., 0.5\%, 0.6\%, 0.8\% on CIFAR-LT-10, 0.7\%, 0.8\%, 0.7\% on CIFAR-LT-100 and 1.1\% on ImageNet-LT. Even on iNaturalist 2018, the performance gain of our method+RIDE over CMO+RIDE is 0.2\%, which is the smallest but still consistent. Besides, ours+ RIDE can achieve better performance in the case of few classes than CMO+RIDE, such as the surprising performance gain of 8.5\%. Moreover, as a data augmentation method, ours can be naturally combined with many existing long-tailed methods or architectures. As summarized in Tables 5 and 6 of Appendix 7.2, we compare ours with CMO when combined with other loss functions, where ours significantly outperforms CMO with various losses. Our OTmix can also be used for balanced classification as a general data augmentation method, while CMO becomes the standard CutMix in the balanced classification task. Ours can be combined with classical mixup-based methods to achieve better performance in balanced classification.
>
> **(2) Response to contribution:** Please see global response.
>
> **(3) Response to visualization:** Following your suggestions, we have provided the visualization of image pair results on ImageNet-LT. As shown in Fig.1 of the pdf uploaded on the ''global'' response, we plot the more semantically reasonable and relevant background image for each foreground image given the learned transport plan. For example, the semantic information of ''Grey fox'' is semantically relevant to ''Dhole''. It visually reveals that our proposed method can create meaningful and advantageous samples, benefiting the balanced and imbalanced classification tasks. We will provide more visualizations in our revision.
>
> **(4) Response to suboptimal solution brought by OT:** We kindly remind you that our proposed method is not finding the closest minority neighbor for a minority sample. Instead, we aim to find the most relevant background image (mainly from majority classes) for each foreground image (mainly from minority classes). Below, we will explain it in detail.
>
> Our key concept is to paste an image mainly from a minority class (foreground) onto rich-context images mainly from a majority class (background). Even though randomly pairing the background and foreground images can generate an acceptable performance in CMO, we find that it might generate more meaningful mixed images if we consider designing the adaptive strategy to pair the foreground and background images. To this end, we propose a more adaptive image-mixing method based on optimal transport. We consider an empirical distribution $P_{\mathrm{back} }$ over the background images and an empirical distribution $Q_\mathrm{fore}$ over the foreground images, where the former are more likely from the majority and the latter from minority classes. To learn the semantic similarity between background and foreground images, we formulate the learning of the similarity as the OT problem between $P_\mathrm{back}$ and $Q_\mathrm{fore}$. In this way, we can view the learned transport plan as the similarity between foreground and background samples, which can be naturally used as guidance to select the most relevant background image for each foreground image.
>
> Besides, similar to your proposed opinion, the classical synthetic minority over-sampling technique (SMOTE) [1] uses the interpolation between a given minority sample and its nearest minority neighbors to create new samples. However, the performance of classical SMOTE is far worse than our baselines, such as Tables (1-2) in M2m [2], where M2m is inferior to ours. It thus indicates that generating the samples between the given minority sample and its nearest minority neighbors is not the ideal solution for long-tailed classification.
>
> [1] SMOTE: Synthetic Minority Over-sampling Technique. [2] M2m: Imbalanced Classification via Major-to-minor Translation.
>
> **(5) Response to the complexity of estimate T:** Thanks for your comments. The optimal transport problem in our paper is computed by the Sinkhorn algorithm [1], which introduces the entropic regularization term for fast computation. To compute the OT distance between $n$ dimensional discrete distributions, the Sinkhorn algorithm requires the computational cost of $\mathcal{O}(n^{2} log(n) /\varepsilon ^{2} )$ reach $\varepsilon$-accuracy. In our case, $n$ corresponds to the batch size, which is usually set to 128 in our experiments.
>
> We have also compared the computational cost of different methods on a Pentium PC with a single GTX 3060 GPU. We report the computational cost (s) per training epoch on long-tailed (Table 7 in Appendix) and balanced datasets (Table 8 in Appendix), respectively.
>
> [1] Sinkhorn distances: Lightspeed computation of optimal transport.

---

> > ### Author Response · Authors · 2023-08-17
> > **Response to Reviewer mXEg**
> >
> > Thank you again for your time. We have provided clarification in our response, which we hope can address your concerns. Please let us know if you have any further concerns. We would appreciate it if you could kindly take our response and other reviewers' comments into consideration.

---

> > > ### Comment · Reviewer_mXEg · 2023-08-17
> > >
> > > I have read authors' response which has addressed my concerns. Thus, I will raise my score to positive.

---

> > > > ### Author Response · Authors · 2023-08-17
> > > >
> > > > Dear reviewer, thanks a lot for your positive feedback to us.

---

### Official Review · Reviewer_HKTJ · 2023-07-05

**Soundness:** 3 good
**Presentation:** 3 good
**Contribution:** 2 fair
**Rating:** 5
**Confidence:** 4

**Summary:**

This paper addresses the interesting long-tailed classification problem, which is usually confronted by real-world machine learning tasks. In particular, this work extends the existing idea of mixing the background images from the majority class and foreground images from the minority class for imbalanced classification by incorporating class-level and sample-level information via the newly developed optimal transport-based image-mixing (OTmix) method. This paper has conducted extensive experiments on artificially crafted and real-world long-tailed datasets, with results demonstrating that OTmix outperforms various types of approaches designed for long-tailed classification, including augmentation-based mechanisms, and cost-sensitive methods, among many others.

**Strengths:**

1. The paper is well-written and presents information in a coherent manner, making it easy to comprehend.
2. The introduced OTmix demonstrates the capability to enhance classification performance on imbalanced datasets. Additionally, it can serve as a valuable data augmentation technique for balanced datasets.
3. The empirical evaluation in this study holds some validity. Extensive experiments have been conducted on diverse long-tailed datasets, and the results provide validation for the superiority of the proposed OTmix compared to state-of-the-art competitors. Furthermore, the ablation studies discussed in Section 5.2 illustrate the importance of each technical contribution made in this research.

**Weaknesses:**

### Major

1. The technical contribution of this research is somewhat constrained. While this study expands on previous methods by incorporating sample-level and class-level information, its technical advancements largely rely on established image-mixing approaches. Hence, it can be considered more of an incremental contribution.
2. This work would benefit from including a baseline comparison with well-known studies in long-tail classification tasks, such as Focal Loss [1] and Class-Balanced Loss [2]. This would provide a more comprehensive evaluation of the proposed approach.
3. The rationale behind leveraging the confusion matrix to learn class-level information is insufficiently explained, making it challenging for readers to comprehend how the confusion matrix can be advantageous for long-tailed classification tasks. Further clarification is needed to elucidate the benefits and implications of utilizing the confusion matrix in this context.



[1] Focal Loss for Dense Object Detection, https://arxiv.org/abs/1708.02002

[2] Class-Balanced Loss Based on Effective Number of Samples, https://arxiv.org/abs/1901.05555

### Minor

1. In P4 L148, it is advisable to provide a concise overview of the proposed framework immediately following "Fig 2." This will aid readers in quickly understanding the significant contributions of this research.
2. Section 5.4 should also conduct explements for exploring the hyperparameter sensitivity of $w$ in Eq (7) and $\lambda$ in Eq (9) on classification performance.

**Questions:**

1. In P3 L130, this paper mentions that "which replaces the image region with a patch from another training image". I wonder what is the patch size used in this paper. Do authors consider the effect of patch size on classification performance? To answer this question, I recommend conducting another ablation study by utilizing various batch sizes. e.g., 4x4, 8x8, or 16x16.
2. How this study works during the inference stage? Should the data in the test set also mix the background and foreground images? If not, could you explain why image-mixing methods can improve the classification performance on the imbalanced dataset?

**Limitations:**

See Weaknesses.

---

> ### Author Rebuttal · Authors · 2023-08-09
>
> **(1) Response to contribution:** Please see global response.
>
> **(2) Response to the comparison with well-known baselines:** Thanks for your suggestion. We provide more comprehensive evaluation experiments with Focal loss and Class-Balanced (CB) loss on long-tailed datasets in the pdf uploaded on the ''global'' response, where Table 3 lists the results of the Focal loss and Table 4 reports the results of the Class-Balanced loss. From the results in Table 3, we observe that Focal + OTmix achieves better performance than Focal and Focal + CMO on CIFAR-LT-10 and CIFAR-LT-100 with different imbalance ratios. Table 4 shows that ours obviously outperforms other methods (CB and CB+CMO) on CIFAR-LT-10, including softmax and sigmoid. These experiments demonstrate the effectiveness of the proposed OTmix in Focal loss and CB loss. We will add these results in our revision if necessary.
>
> **(3) Response to the reason for using the confusion matrix:** Thanks for your comments. This work aims to learn the optimal transport plan between the background images and foreground images. The cost function $C_{ij}$ measures the distance between the $i$-th background image and the $j$-th foreground image, which plays an important role in learning the optimal transport plan. Generally, the cost function $C_{ij}$ should be small if the $i$-th background image and $j$-th foreground image are close, and a small cost function $C_{ij}$ can result in a large transport plan element $T_{ij}$, where the image $i$ is thus more likely to mix with the image $j$. Therefore, we need to design a proper cost function. A simple but straightforward way is defining the sample-level function using the features of $x_i$ and $x_j$. However, this way ignores the class information of $x_i$ and $x_j$. For example, it is reasonable that the class about cat is close to the class about dog than the class about airplane. Therefore, we set the diagonal element as 0 to achieve the normalized confusion matrix, whose element $\hat{F}\_{y_i, y_j}$ shows the class similarity between  $y_i$ and $y_j$. After that, we define the cost function as $C_{ij} =1-\hat{F}\_{y_i, y_j}$.  When the class of $y_i$ is most easily confused with the class of $y_j$ in the confusion matrix, $C_{ij}$ would be small and $T_{ij}$ would be large, where we are inclined to mix the image $x_i$ and image $x_j$. So designing the class-level cost function using the confusion matrix is beneficial for us to implement the image-mixing process.
>
> **(4) Response to the concise overview of the proposed framework:** Following your suggestion, we provide a concise overview of the proposed framework as below. We consider a distribution $P_\mathrm{back}$ over the background images mainly from majority classes and another distribution $Q_\mathrm{fore}$ over the foreground images mainly from minority classes. By minimizing the OT distance between these two distributions, where we design the sample-level and class-level cost function, we use the learned transport plan as guidance to create the mixed images between the two distributions.
>
> **(5) Response to the hyperparameter sensitivity of $\omega $ and $\lambda$:** Thanks for your comments. The sensitivity of the hyperparameter $\omega $ in Eq (7) on classification performance has been discussed in Fig.6 (a) in Appendix 7.3. In terms of the combination ratio $\lambda$ in Eq (9), it is sampled from the beta distribution $\mathrm{Beta} (\alpha, \alpha)$ (see line 133 in our submitted version). We have discussed the effect of the hyperparameter $\alpha$ in Fig.6 (b) in Appendix 7.3. Besides, we also explore the experiments about the hyperparameter $r$ and the training factor $\mu $ in Fig.6 (c) and Fig.6 (d) in Appendix 7.3, respectively. We will make it more clear in our revision.
>
> **(6) Response to patch size:** Thanks for your comments. We kindly remind you that we do not use a fixed patch size (i.e., the size of the bounding box) during the training stage, where we randomly sample the bounding box following the Cutmix in each mini-batch.
>
> In Appendix 7.1, we have provided the details of the bounding box, which is calculated in Eq (11). Specifically, the bounding box coordinate is $\mathbf{B}=\left(r_x, r_y, r_w, r_h\right)$. Here, $r_{x}$ and $r_{y}$ are randomly sampled from the uniform distribution, $r_{w}$ and $r_{h}$ depend on the combination ration $\lambda $, where $\lambda $ is sampled from the beta distribution $\mathrm{Beta} (\alpha, \alpha)$. Therefore, different bounding boxes can be randomly generated to improve sample diversity in each mini-batch. Besides, we have performed the ablation study about the hyperparameter $\alpha$ in Fig.6 (b) in Appendix 7.3.
>
> **(7) Response to the test image:** Thanks for your comments. During the inference stage, we do not need to mix the background and foreground images. Following the standard classification pipeline, the sample in the test set will be directly fed into the feature extractor followed by the classifier to predict the label of the input. That is to say, we only mix the images in the training stage based on OTmix. It provides an adaptive way to learn the similarity between background and foreground images. Now, the rich context of the majority classes as background images can be better transferred to the minority classes as foreground images. Since we augment more semantically reasonable and meaningful mixed images to train the classifier, OTmix improves the classification performance on the imbalanced dataset. Extensive experiments also demonstrate the effectiveness of our method for long-tailed classification.

---

> > ### Comment · Reviewer_HKTJ · 2023-08-14
> > **Responses to the Rebuttal**
> >
> > We thank the authors for their efforts in addressing my concerns, especially conducting additional experiments by using the Focal Loss and CB Loss in Tables 3 and 4 of the rebuttal pdf. After carefully reading other reviewers' comments and the authors' responses, my concern is that the use of the confusion matrix hinders the generability of the proposed method for large-scale long-tailed datasets. For example, iNaturalist 2018 has a total of 8,142 classes, requiring a confusion matrix of 8142x8142 dimensions, which significantly increases the computational overhead. But, except for the confusion matrix, this paper is good and slightly beyond the accepted threshold. Therefore, I would like to keep my original score.

---

> > > ### Author Response · Authors · 2023-08-18
> > >
> > > Thank you for your response. We would like to explain that computing the confusion matrix does not add much computational overhead to the training of our method. Specifically, in each epoch, the confusion matrix is computed only once. Moreover, the confusion matrix does not have to be computed over the whole training set and we can sample a balanced subset instead. On iNaturalist 2018, one training epoch of our method with the confusion matrix takes about 1,300 seconds, among which, the computation of the confusion matrix only takes about 86 seconds. We appreciate the useful comment and will study more on this point in the revision.

---

### Official Review · Reviewer_YYXw · 2023-07-06

**Soundness:** 3 good
**Presentation:** 3 good
**Contribution:** 3 good
**Rating:** 6
**Confidence:** 4

**Summary:**

The authors tackle the notable challenge of learning from imbalanced data, proposing an adaptation to mixup-based or copy-paste-based data augmentation for rare categories that takes sample-level semantic similarity into account. Their proposed method uses optimal transport to underpin an adaptive mixup strategy, combining sample- and image-level information. They show this context-aware mixup strategy is useful as data augmentation for both long-tailed and balanced datasets.

After reading the authors rebuttal and discussing with them, I will maintain my score of 6.

**Strengths:**

The method is intuitively motivated, the implementation is reasonably simple, and the results are good and demonstrated on real-world imbalanced datasets such as iNaturalist. The authors further demonstrate improvement over vanilla mixup for balanced datasets like CIFAR.

**Weaknesses:**

Though they emphasize the semantic quality improvement of their method, it’s clear that this could perhaps be taken even further. For example, the method places pandas (a bear) on top of images of other bears, which is reasonable at a surface level. But these bears have drastically different habitats, at a more nuanced level it’s fundamentally unreasonable to see a panda and a black bear in the same habitat.

Performance is significantly worse on common categories in iNaturalist when using this method than ERM+CMO, and as common classes are the vast majority of what is seen in the real world, the drop in overall accuracy of this method when deployed on imbalanced test data might be difficult to overcome.

Nits:

Grammar is often incorrect throughout, too many small errors to reasonably capture here

It would be good to reference https://arxiv.org/abs/2302.11861, which shows targeted copy-paste augmentation aligned with a novel test domain can improve out of domain robustness, a similar finding, as well as https://arxiv.org/abs/1904.05916 which shows rare-class performance improvement from synthetic data including a copy-paste baseline which includes context of night vs day backgrounds.

**Questions:**

Bias and imbalance is often as related to where data is captured as it is to what is captured. How might bias in image backgrounds towards specific parts of the world, for example, impact the effectiveness of this type of approach?

**Limitations:**

Little to no discussion of limitations or failure modes of this method, it would improve the quality of the work to understand where this helps and where it does not. Does performance for any rare category get worse?

---

> ### Author Rebuttal · Authors · 2023-08-09
>
> **(1) Response to unreasonable samples:** Thanks for your insightful comments. We agree that for any imperfect method, there can be cases where foreground patches are mixed unreasonably with background patches, like the example you pointed out. We would like to point out that CMO mixes patches completely randomly while ours does it guided by an adaptive OT transport plan. Therefore, we believe that the overall proportion of unreasonable mixed images by ours should be less than that of CMO, which explains that ours outperforms CMO. To empirically verify this, we leverage the method of data valuation[1], which assesses the quality of data to see whether a training dataset can improve model performance or mitigate undesirable biases. Specifically, we generate 512 mixed samples as training data and randomly select 128 images from the test split of CIFAR-LT-100 as test data by OTmix and CMO, respectively. We then use the data valuation algorithm in [2] implemented in pyDVL to examine the quality of the mixed images of the two methods. The comparison result of the average data values (larger is better) is OTmix=0.198 and CMO=0.115. It indicates that OTmix generates higher-quality mixed images than CMO. In the emerging area of using mixed images for long-tailed problems, we agree that there is surely room for further improvement, which will be studied in our future work.
>
> [1] OpenDataVal: a Unified Benchmark for Data Valuation. [2] Data Shapley: Equitable Valuation of Data for Machine Learning
>
> **(2) Response to the performance:** We agree that our performance on common (Many) categories is not as good as ERM+CMO's on iNaturalist. However, we have significant improvements in overall, medium, and few categories on CIFAR-LT-10, CIFAR-LT-100, ImageNet-LT, and iNaturalist 2018 datasets. These experiments demonstrate the overall effectiveness of our OTmix for imbalanced classification.
>
> **(3) Typos:** We will carefully address the grammar issues in our revision.
>
> **(4) Response to references:** Thanks for your references, which are quite inspiring. [1] proposes a targeted augmentation, Copy-Paste (Same Y), which randomizes backgrounds between cameras in similar habitats, where the relevant dataset to long-tailed datasets is species classification from camera trap images (IWILDCAM2020-WILDS). [2] proposes CCT-20 dataset come from the Caltech Camera Traps (CCT) dataset, where cis-locations are defined as cations seen during training and trans-locations as locations not seen during training. Both papers are used to solve multi-domain unbalanced classification problems, which are related but also differ from our task, the single-domain unbalanced classification problem. The dataset in [1] is quite large and [2] does not disclose the code or dataset. Therefore, it would be a bit hard for us to conduct experiments to compare with them in the rebuttal phase. We will put them into our reference and provide a more detailed study in our revision.
>
> [1] Out-of-Domain Robustness via Targeted Augmentations. [2] Synthetic Examples Improve Generalization for Rare Classes.
>
> **(5) Response to bias and imbalance:** To empirically explore the impact of bias in image backgrounds towards specific parts of the world on our method, we conduct an additional experiment of classification accuracy in the group-oblivious setting of ResNet-50 on Waterbirds[1] in Table 2 in the pdf uploaded on the ''global'' response.
>
> **Dataset.** Waterbirds is a biased dataset in image backgrounds, whose task is to distinguish whether the bird is a waterbird or a landbird. According to the background and label of an image, this dataset has four predefined subpopulations, i.e., ''landbirds on land (Group 0)'', ''landbirds on water (Group 1)'', ''waterbirds on land (Group 2)'', and ''waterbirds on water (Group 3)''. In the training set, the largest subpopulation is Group 0 with 3,498 samples, while the smallest subpopulation is Group 2 with only 56 samples. We emphasize that the training set is biased, whereas the validation set and the test set are balanced.
>
> **Evaluation metrics.** Following [2], we report the average accuracy and worst-group (group 2) accuracy, with a focus on the worst-case accuracy, which is more important than the average accuracy in some application scenarios ad stated by [2].
>
> **Baseline Methods.** We consider several baseline methods: (1) ERM (the cross-entropy loss); (2) CMO; (3) Focal loss, which weights well-classified samples; (4) Methods of distributionally robust optimization (DRO) for group shifts: $\chi ^{2}\mathrm{-DRO}$[3] and $\chi ^{2}\mathrm{-DORO}$[4].
>
> **Results.** Table 2 demonstrates that ours achieves a desired trade-off between the average and worst-case accuracy, a well-known challenge [2]. Besides, the worst-group accuracy of ERM+OTmix outperforms other competing methods. Specifically, ERM+OTmix improves 18.3\% about the worst-group accuracy than ERM+CMO and 3.4\% than $\chi ^{2}\mathrm{-DORO}$. Although ERM+OTmix about the average accuracy is 1.1\% lower than ERM, the performance of other methods also declined, where ours outperforms Focal loss and two methods of DRO. These results illustrate that  OTmix can still perform effectively on the biased dataset.
>
> [1] Distributionally robust neural networks.
> [2] UMIX: Improving Importance Weighting for Subpopulation Shift via Uncertainty-Aware Mixup. [3] Large-scale methods for distributionally robust optimization.
> [4] Doro: Distributional and outlier robust optimization.
>
> **(6) Response to limitations:** We have discussed negative societal impacts and limitations in Appendix 7.6. The imprecise or incorrect confusion matrix can mislead our method to build unsatisfactory pairs for image-mixing and to generate invaluable minority samples. So we need to provide a trained model or train a model from the beginning until the confusion matrix can be sufficient to provide the sample-level information for our method. It may spend some time.

---

> > ### Comment · Reviewer_YYXw · 2023-08-19
> > **Clarification**
> >
> > I want to clarify for the authors that in (4) I would not expect additional experiments. As the authors pointed out, the multi-domain setting is different from their study setting. I included these references as I thought they may provide additional context or motivation in the related work, as in both cases the semantic relevance of the combination of background and foreground for data augmentation is important to improved performance.
> >
> > I appreciate the authors responses to my questions and clarifications. I think it would be worth refactoring the manuscript to include the discussion of the confusion matrix quality as a necessary condition for the method to work well, as this is an interesting and important insight into their method's requirements for use. Do the authors think it would be possible to include a more detailed discussion of this in the manuscript?
> >
> > Based on the authors response and the other reviews, I am inclined to maintain my positive score.

---

> > > ### Author Response · Authors · 2023-08-19
> > >
> > > Thank you for your response and insightful comment. We will cite these two references and provide a detailed study on the confusion matrix quality in our revision, as suggested.

---

### Official Review · Reviewer_ARR2 · 2023-07-06

**Soundness:** 3 good
**Presentation:** 4 excellent
**Contribution:** 3 good
**Rating:** 7
**Confidence:** 5

**Summary:**

To tackle the data imbalance problem,  the authors suggest to manipulate the data distribution by introducing a new data augmentation mechanism.  This mechanism is a guided mix-up operation that uses backgrounds mostly from major classes and foregrounds from low frequency classes. To make the augmented images look more reasonnable, the authors suggest to match images based on a similarity at sample and/or class level. They formulate the  matching problem as a transport problem. The performance of the introduced augmentation is validated by several experiments on different benchmarks as well as balanced datasets.

**Strengths:**

Although this paper introduces a simple extension of the mixup operation, the intuition behind it makes it an elegant solution. The formulation is technically sound and the claims are sufficiently supported with experiments. The paper is also well-written.

**Weaknesses:**

minor weakness: some typos that can be fixed easily

**Questions:**

Did the authors also consider to keep a fraction of the images unchanged in every batch? I am wondering if the trained networks will perform better in this case on real images than if they have been exclusively trained on mixed images

---

> ### Author Rebuttal · Authors · 2023-08-09
>
> **(1) Response to typos:** Thanks for your reminder and we will carefully address them in the revised version.
>
> **(2) Response to the fraction about the mixed images:** According to your suggestion, we conduct an additional experiment with the fraction of the images unchanged in every batch in the pdf uploaded on the ''global'' response. We denote $Fraction = \frac{Mixed}{Overall}$ in every batch. From the results in Table 1, when the number of mixed samples decreases ($Fraction\downarrow $), the performance of OTmix deteriorates significantly, which suggests that it is better to use mixed samples alone in every batch.

---

> ### Comment · Area_Chair_S6w1 · 2023-08-19
> **Reminder -- please reply to author comments**
>
> Dear Reviewer ARR2,
>
> Thank you again for reviewing this paper. Since the discussion with authors is closing soon, could you please respond to the rebuttal?
>
> Best,
>
> AC

---

### Author Rebuttal · Authors · 2023-08-09

We thank all reviewers for their constructive comments. We upload a global pdf about the figures and tables. Below we first address common concerns and then respond to each reviewer.

**Response to Reviewers HKTJ and mXEg about contribution and novelty:** Learning relatively balanced classes from the data perspective is an effective solution to the long-tailed problem, where the study on mixup-based methods has gradually attracted the attention of the researchers. Among them, CMO recently constructs mixed samples between the background images more likely from the majority classes, and foreground images more likely from the minority classes. Despite the effectiveness of CMO, it may generate unreasonable mixed samples for arbitrarily pairing the foreground and background images and ignoring their semantic distances. Therefore, designing an effective method to pair the background images and foreground images is quite necessary for creating reasonable augmented samples.

To this end, we propose a more adaptive image-mixing method for generating semantically reasonable, and meaningful mixed samples based on optimal transport (OT) on long-tailed classification. It is non-trivial since we formulate the image-mixing problem as the problem of learning the optimal transport plan between two discrete empirical distributions, where one distribution is over the background images and another is over the foreground images. Due to the importance of the cost function in learning the transport plan for OT, we further design the cost function using the sample-level representation and class-level label information. Once we obtain the transport plan, we use it as the similarity guidance to select the most relevant background image for each foreground image. Besides, our proposed method can also be adapted to the balanced classification setting, where CMO becomes the standard Cutmix. It shows that our proposed method can serve as a general data augmentation method, which is effective in both imbalanced classification and balanced classification.

To summarize, CMO is the state-of-the-art method in the line of mixup-based methods for long-tailed classification. We propose a new method in this line that improves over CMO via nontrivial efforts, which we believe has significant contributions.

---

### Decision · Program_Chairs · 2023-09-21

**Decision:**

Accept (poster)

**Comment:**

This paper proposes an adaptive image-mixing augmentation method to generate semantically meaningful images of minority classes for long-tailed classification. The proposed method employs optimal transport (OT) to incorporate class-level and sample-level information for adaptive mixing. Initially, reviewers raised concerns such as the generated image may be unreasonable, performance drops on common categories, the contribution is incremental, the performance gain is limited, lack of comparisons with some baselines, and the rationale of using the confusion matrix and OT. Although not all concerns were well addressed in the rebuttal, all reviewers recommended positive final ratings. Considering that the method is intuitively motivated, technically sound, and effective, as appreciated by reviewers, the AC follows this unanimous recommendation. Reviewers did raise valuable concerns that should be addressed. The authors are encouraged to make the necessary changes in the camera-ready version.